# A COVID-19 Patient with Simultaneous Renal Infarct, Splenic Infarct and Aortic Thrombosis during the Severe Disease

**DOI:** 10.3390/healthcare10010150

**Published:** 2022-01-13

**Authors:** Georgios Mavraganis, Sofia Ioannou, Anastasios Kallianos, Gianna Rentziou, Georgia Trakada

**Affiliations:** Department of Clinical Therapeutics, Alexandra Hospital, Medical School, National and Kapodistrian University of Athens, P.O. Box 11528 Athens, Greece; giomavraganis@gmail.com (G.M.); sofioan7@gmail.com (S.I.); kallianos.pulm@gmail.com (A.K.); giannarentziou@gmail.com (G.R.)

**Keywords:** COVID-19, thrombotic complications, renal infarct, splenic infarct, aortic thrombosis

## Abstract

Severe acute respiratory syndrome coronavirus 2 (SARS-CoV-2) infection has been associated with a high incidence of arterial and venous thrombotic complications. However, thromboembolic events in unusual sites such as limb and visceral arterial ischemia are reported rarely in the literature. Herein, we describe a rare case of a patient with severe coronavirus disease 2019 (COVID-19) infection who experienced severe abdominal pain during the hospitalization and presented simultaneously renal artery, splenic artery and vein as well as aortic thrombi despite prophylactic antithrombotic treatment. Information about his follow-up post discharge is also provided. This case report raises significant clinical implications regarding the correct dose of antithrombotic treatment during the acute phase of the severe COVID-19 infection and highlights the need for incessant vigilance in order to detect thrombosis at unusual sites as a possible diagnosis when severe abdominal pain is present in severe COVID-19 patients.

## 1. Introduction

Severe acute respiratory syndrome coronavirus 2 (SARS-CoV-2) infection has been associated with a high incidence of arterial and venous thrombotic complications [1]. Venous thrombotic events, mainly deep venous thrombosis (DVT) and pulmonary embolism, present with higher frequency followed by arterial thromboembolic events including acute coronary syndromes and ischemic strokes, especially in severe coronavirus disease 2019 (COVID-19) patients [2]. However, arterial thromboembolic events have been rarely reported in unusual sites such as limb and visceral arterial ischemia [2]. This raises questions over the appropriate dose of anticoagulants in severe COVID-19 patients, since numerous thrombotic events occur despite the administration of prophylactic dose of antithrombotic treatment [1].

Herein, we report the case of a 64-year-old man with severe COVID-19 infection who presented renal artery, splenic artery and vein thrombosis as well as aortic thrombi during his hospitalization along with his imaging and clinical-follow up for one month post discharge.

## 2. Case Description

A 64-year-old patient, unvaccinated for SARS-CoV-2, body weight 94 kg, without known medical history of comorbidities, was admitted to the emergency department for fever up to 38.7 °C for two weeks in addition to shortness of breath and dry cough for 3 days accompanied with desaturation (oxygen saturation 80%) at the last day.

On admission, the patient was febrile (38.5 °C), with type 1 respiratory failure, oxygen saturation 86% and the clinical examination revealed decreased sounds across all pulmonary fields along with a breathing rate of 30 per minute. The initial chest X-ray revealed bilateral lung infiltrates covering over 50% of both lungs. The patient underwent polymerase chain reaction test for SARS-CoV-2 that was positive and was admitted in a COVID-19 ward. The laboratory tests were as follows: lymphopenia (950 lymphocytes), serum levels of alanine and aspartate aminotransferases of 130 U/L (normal range <40 U/L) and 135 U/L (normal range <40 U/L), respectively, plasma d-dimers of 0.57 mg/L (<0.5 mg/L), high-sensitivity C-reactive protein (hs-CRP) 19.9 mg/dL (normal range <0.5 mg/dL), creatine kinase (CPK) of 1215 U/L (normal range <170 U/L) and lactate dehydrogenase (LDH) of 551 U/L (normal range <225 U/L). The patient was treated with low molecular heparin at prophylactic dose (enoxaparine 6.000IU once daily), intravenous dexamethasone 6 mg and remdesivir once daily for 5 days, omeprazole, tocilizumab 800 mg (one dose) and ceftaroline 600 mg (two doses per day). Oxygen therapy using non-rebreather mask was immediately supplied and on day 2, the patent required high-flow nasal cannula (60 L/min, 90% FiO2) for his oxygen demands and acute respiratory distress syndrome (ARDS).

On day 5, the patient experienced referred abdominal pain, localized mostly on the left upper and lower quadrant.

The blood examinations showed leukocytosis (20,200 white blood cells (WBCs) with 19,300 polymorphonuclear cells), LDH of 1244 U/L and plasma d-dimers of 3.7 mg/L while high-sensitivity C-reactive protein was 1.7 mg/dL. The patient underwent immediately computed tomography (CT) of the abdomen and pelvis as well as CT pulmonary angiography (CTPA). The latter showed bilateral ground glass opacities and regions with pulmonary consolidation but no evidence of pulmonary embolism (Figure 1). Notably, thrombi up to 5 mm were detected in the thoracic aorta. Meanwhile, the abdominal CT scan showed wedge-shaped hypodensities in the lower lobe of the left kidney consistent with renal infarct (Figure 2). Furthermore, peripheral brunches of the left renal artery as well as of splenic artery and vein were not well attenuated while, at the same time, a large area of hypoattenuation in the splenic parenchyma was detected, consistent with splenic infarct (Figure 2).

The patient was then administered enoxaparine at therapeutic dose (8.000 IU twice daily) along with acetylsalicylic acid 80 mg one time per day. On day 8, enoxaparine was replaced with subcutaneous (sc) fondaparinux 7.5 mg. In the context of the diagnostic algorithm for thrombotic complications, the patient underwent serologic blood exams for antiphospholipid syndrome and Janus kinase 2 mutations, which were all negative. The patient gradually improved and, on day 11, high-flow nasal cannula was replaced with a Venturi mask (FiO2 50%) for oxygen supply. At the same time, the biochemical and hematologic parameters were gradually improved (WBCs 16,000, d-dimers 3.2 mg/L, LDH 894 U/L). Dexamethasone continued to be administered and was ceased at day 18, after appropriate tapering. On day 19, the patient was discharged from hospital, free from any symptoms.

### Clinical Follow-Up

The patient was instructed to receive amoxicillin per os twice daily at the dose of 500 mg along with acetylsalicylic acid 80 mg, sc fondaparinux 7.5 mg and omeprazole 20 mg per os for six months. Additionally, the patient was advised to repeat the blood examinations along with angiography at one month. He also underwent duplex ultrasound of carotid and femoral arteries and lower extremity venous ultrasound which both showed no presence of thrombus or stenoses or significant atheromatous plaques. The patient also repeated the abdominal and pelvic CT scan one month after the thrombosis at the acute phase, which again demonstrated the presence of significant splenic infarct with only the lower pole of the spleen remaining functional and the presence of infarct at the lower lobe of the left kidney along with renal cortical thinning. Lung CT scan showed no abnormalities regarding thrombotic complications. The blood exams performed at one-month post-hospital discharge showed normal count of WBCs (10,200), hematocrit 46.2% and hemoglobin 14.8 g/dL; albeit, Howell-Jolly bodies were detected signifying functional asplenia so the patient was advised to vaccinate against encapsulated bacteria; until this, he was advised to continue amoxicillin as chemoprophylaxis. The timeline of the clinical sequelae of the patient, including both hospitalization and clinical follow-up, is briefly reported in Table 1.

## 3. Discussion

To our knowledge, this is the first case report demonstrating the simultaneous presence of both venous and arterial thrombotic complications involving thoracic aorta, left renal artery and both splenic artery and vein, occurring at the acute phase of COVID-19 infection in a hospitalized patient in combination with information about his clinical follow-up. Notably, renal and splenic infarct were still detected one-month post-discharge indicating their persistent effect as long-term sequelae of COVID-19 infection.

COVID-19 has been strongly associated with an increased incidence of thrombotic complications, both arterial and venous, including events such as DVT, pulmonary embolism, acute coronary syndromes and ischemic strokes due to its hypercoagulable state and systemic inflammation which raise substantially the risk within the first week of illness [3,4]. In detail, several coagulation abnormalities predisposing to thrombosis such as longer prothrombin and activated partial thromboplastin times, elevated D-Dimer levels and fibrin degradation products as well as severe thrombocytopenia possibly leading to disseminated intravascular coagulation have been reported in COVID-19, highlighting the need for incessant vigilance [5]. Moreover, the hyperinflammatory response commonly observed in COVID-19 may also cause endothelial inflammation and microvascular damage leading to heightened risk of thrombotic events along with the excessive venous stasis due to immobilization and platelet activation in hospitalized patients [6,7]. In this context, clinicians should be highly aware of potential thrombotic complications and close monitoring of coagulation abnormalities should be prioritized.

The SARS-CoV-2 may affect kidney function through both direct effect of SARS-CoV-2 on kidney tubular and endothelial cells and indirect damage by release of cytokines, in addition to potential hypoperfusion due to restrictive fluid strategy [8,9,10], while cases of kidney infarction in hospitalized COVID-19 patients have been also described, indicating the effect of COVID-19 on thrombotic complications observed in the renal system [11,12]. However, the non-specific clinical presentation of renal infarct including abdominal or flank pain, nausea, fever or hematuria along with the low prevalence of this condition ranging 0.1 to 1.4% makes the diagnosis challenging [13,14,15]. Furthermore, splenic infarcts have also been reported in patients hospitalized with COVID-19 [16]. Similarly, splenic thrombosis is infrequent and should be suspected in patients with left-sided abdominal pain [17]; however, due to indistinct clinical presentation, the incidence of splenic infarcts is often underestimated as abdominal imaging is not routinely performed and they are often incidental findings. However, as also shown in our case, the presence of Howell–Jolly bodies certified the splenic dysfunction, as this type of red blood cell would normally be filtered by a functional spleen [18]. Importantly, our case demonstrated that the functional asplenia may persist even months after the acute phase of splenic infarct highlighting the need for continuous antibiotic treatment, correct administration of vaccines and vigilance for potential infections. Aortic thrombosis, another complication rarely seen in the acute COVID-19 sequelae [7,19,20] was also reported at our patient and was attributed to severe COVID-19 pneumonia inducing prothrombotic state through one of the aforementioned mechanisms. Notably, antiphospholipid antibodies frequently seen in other cases with arterial thrombotic complications were not detected in our patient, nor were anticardiolipin antibodies that could partially explain the acquired coagulopathy in this patient [21,22].

This case raises significant clinical implications regarding the appropriate dose of anticoagulant treatment (prophylactic or therapeutic) in severe COVID-19 patients and highlight the need for incessant vigilance for thrombotic complications in hospitalized patients. First, the symptom of severe abdominal pain should include visceral infarction in its differential diagnosis and appropriate imaging and laboratory diagnostic work-up is essential to rule out acute vascular thrombosis in this setting. Second, in patients with acute thrombosis during hospitalization, not only the allocation of appropriate anticoagulant treatment regimen but also at correct dose may improve prognosis, alleviate symptoms and significantly reduce mortality rates and future disability. Third, in these patients, clinical and imaging follow-up post-discharge is essential in order to assess the viability of organs and allocate correct chronic anticoagulant and other treatment strategies.

## 4. Conclusions

In conclusion, this case demonstrates the simultaneous presence of arterial and venous thromboses in a patient hospitalized with severe COVID-19 pneumonia which could persist for months post-discharge and stress the importance for ceaseless surveillance of clinicians regarding these complications, not only in the acute phase but also at the long-term sequelae of COVID-19 infection.

## Figures and Tables

**Figure 1 healthcare-10-00150-f001:**
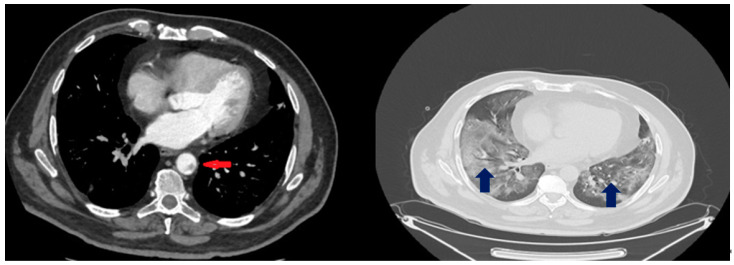
CT pulmonary scan of the patient at day 6. The blue arrows demonstrate bilateral ground glass opacities and regions with pulmonary consolidation. The red arrow indicates the thrombi detected in the thoracic aorta.

**Figure 2 healthcare-10-00150-f002:**
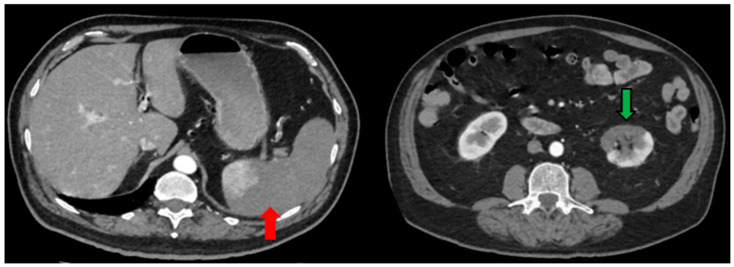
Abdominal CT scan of the patient at day 6. The red arrow demonstrates the area of hypoattenuation in the splenic parenchyma consistent with the splenic infarct. The green arrow indicates wedge-shaped parenchyma hypodensities in the lower lobe of the left kidney, typically seen in renal infarcts.

**Table 1 healthcare-10-00150-t001:** Timeline of clinical sequelae of the patient.

Time	Event
Day 0 (hospital admission)	Fever 38.5 °C, breath shortness, dry cough, oxygen saturation 86% PCR test for SARS-CoV-2 infection: positive Blood exams: 950 lymphocytes, d-dimers: 0.57 mg/L, LDH 551 U/L Enoxaparine at prophylactic dose (6.000IU once daily) Non-rebreather mask
Day 2	HFNC due to clinical deterioration and presence of ARDS
Day 5	Blood exams: leukocytosis (20,200 WBCs with 19,300 polmorphonuclear cells), LDH 1244 U/L, d-dimers 3.7 mg/L. Abdominal pain, localized mostly on the left upper and lower quadrant
Day 6	Blood exams: 23,000 WBCs, LDH 1892 U/L, d-dimers 4.3 mg/L CTPA: Bilateral ground glass opacities, no evidence of pulmonary embolism, thrombi up to 5 mm in the thoracic aorta Abdominal and pelvic CT scan: Splenic infarct, thrombosis of splenic artery and vein, renal infarct at the left lower lobe. Enoxaparine at therapeutic dose (8.000 IU twice daily), acetylsalicylic acid 80 mg
Day 8	Enoxaparine was replaced to sc fondaparinux 7.5 mg × 1
Day 11	Venturi mask (FiO2 50%), clinical improvement Continuing treatment with dexamethasone
Day 18	Dexamethasone ceased after appropriate tapering Nasal oxygen therapy at 2 L/min
Day 19	Patient without oxygen demands Discharge from hospital with the following instructions: po amoxicillin 500 mg × 2, po acetylsalicylic acid 80 mg, sc fondaparinux 7.5 mg and po omeprazole 20 mg for 6 months
Day 37	Ultrasound of femoral and carotid arteries: No presence of thrombus, stenosis or significant atherosclerotic plaques
Day 43	CTPA: Residual ground-glass opacity lesions, fibroatelectacic lesions at the right lower and middle lobe, traction bronchiectasis at right middle lobe and lingula of left lobe. No presence of thrombi at the thoracic aorta. Abdominal and pelvic CT scan: Splenic and renal infarct (along with renal cortical thinning) are depicted again. Re-tunelling of the splenic vein.
Day 51	Ultrasonography of the lower extremities’ veins: No detection of venous chronic insufficiency or DVT. Blood exams: 10,200 WBCs, hematocrit 46.2% and hemoglobin 14.8 g/dL, detection of Howell-Jolly bodies Patient instructed to receive pneumococcal 13-valent conjugate vaccine, haemophilus influenza type b vaccine (Hib-ActHIB) as well as the meningococcal vaccine.

Abbreviations: PCR, polymerase chain reaction; SARS-CoV-2, severe acute respiratory syndrome coronavirus 2; LDH, lactate dehydrogenase; HFNC, high-flow nasal cannula; ARDS, acute respiratory distress syndrome; WBCs, white blood cells; CTPA, computed tomography pulmonary angiogram; CT, computed tomography; po, per os; sc, subcutaneous; DVT, deep vein thrombosis; ActHIB, haemophilus b conjugate vaccine.

## Data Availability

Data sharing not applicable.

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
