# Peer review of "A COVID-19 Patient with Simultaneous Renal Infarct, Splenic Infarct and Aortic Thrombosis during the Severe Disease"

_healthcare, 2022, doi:10.3390/healthcare10010150_

Round 1

Reviewer 1 Report

Dear Authors,

Your paper is very well written and presents an important case.
There are very few issues in the paper:

  • Is your remark at lines 115 – 116 vulnerable: ‘This is the first case reporting …”?

    This important remark needs a brief check against references [1 – 3]

  • Below I give recommendations:

Questions, suggestions, and discussions:

  1. Page 3, line 100 and page 4 ‘Day 5’: the spelling ‘Howell-  Jolly’ is wrong. The blank before the J should be deleted;

  2. Page 4, lines 115-118: is it correct that you neglected reference to the venous and arterial complications published from two other European countries, recently: Lodigiani [1], Levolger [2], Klok [3]? Currently, you only refer in line 125 to Avila, i.e. ref. [3];

  3. Page 6, lines 186 – 187: missing page numbers in ref. [2];
    - line 149, should the reference to Nakagami [17] be extended with ref. to Alisjahbana  [4] below?
              - Table 1, lines 106 - 106: inconsistent layout: ‘Blood exams’ are written bold, or italic and bold, or not any of these (Day 5). Why is ‘Traction’ at the second line of page 4 written with a capital ‘T’ ? and do the final sentences in both Days 43, 51 end with a dot?

    Please check the content of Table 1 for readability;

Additional/Missing References

  1. Lodigiani, C.; Iapichino, G.; Carenzo, L.; Cecconi, M.; Ferrazzi, P.; Sebastian, T.; Kucher, N.; Studt, J.D.; Sacco, C.; Alexia, B.; et al. Venous and arterial thromboembolic complications in COVID-19 patients admitted to an academic hospital in Milan, Italy. Thromb. Res. 2020, 191, 9–14.
  2. Levolger, S.; Bokkers, R.P.H.; Wille, J.; Kropman, R.H.J.; de Vries, J.P.P.M. Arterial thrombotic complications in COVID-19 patients. J. Vasc. Surg. Cases, Innov. Tech. 2020, 6, 454–459.
  3. Klok, F.A.; Kruip, M.J.H.A.; van der Meer, N.J.M.; Arbous, M.S.; Gommers, D.A.M.P.J.; Kant, K.M.; Kaptein, F.H.J.; van Paassen, J.; Stals, M.A.M.; Huisman, M. V.; et al. Incidence of thrombotic complications in critically ill ICU patients with COVID-19. Thromb. Res. 2020, 191, 145.
  4. Alisjahbana, B.; Huang, I.; Oehadian, A. The detection of Howell-Jolly body-like inclusions in a case of coronavirus disease-2019 (COVID-19). Blood Res. 2020, 55, 191.

Author Response

  • Is your remark at lines 115 – 116 vulnerable: ‘This is the first case reporting …”?
    This important remark needs a brief check against references [1 – 3]

We added the word “simultaneous” to show that our case is the first presenting the simultaneous presence of both venous and arterial thrombotic complications involving thoracic aorta, left renal artery and both splenic artery and vein.

  • Below I give recommendations:

Questions, suggestions, and discussions:

  1. Page 3, line 100 and page 4 ‘Day 5’: the spelling ‘Howell-Jolly’ is wrong. The blank before the J should be deleted;
    We corrected it.
  2. Page 4, lines 115-118: is it correct that you neglected reference to the venous and arterial complications published from two other European countries, recently: Lodigiani [1], Levolger [2], Klok [3]? Currently, you only refer in line 125 to Avila, i.e. ref. [3];
    We added Lodigiani et al.
  3. Page 6, lines 186 – 187: missing page numbers in ref. [2]; We corrected it.
    - line 149, should the reference to Nakagami [17] be extended with ref. to Alisjahbana  [4] below?
    Alisjahbana et al presented a case report from a COVID-19 patient with Howell-Jolly.

          - Table 1, lines 106 - 106: inconsistent layout: ‘Blood exams’ are written bold, or italic and bold, or not any of these (Day 5). Why is ‘Traction’ at the second line of page 4 written with a capital ‘T’? and do the final sentences in both Days 43, 51 end with a dot?
We corrected it.

Reviewer 2 Report

The article presents a case report with the simultaneous development of thrombosis of the renal artery, splenic artery and vein, as well as thrombosis of the aorta, despite prophylactic antithrombotic treatment in severe COVID-19 infection. This case is of interest to medical practitioners.

Author Response

There was not a comment by the reviewers that needed to be addressed.

Reviewer 3 Report

Couple of queries: did not see if an echocardiogram was done; if so please comment; was there a PFO? Also were the valves normal without any pathology?

This was a 64year old patient; do we know about the status of his coronaries; lipid profile? Were there any atheromatous plaques in the descending thoracic/abdominal aorta?

Author Response

Couple of queries: did not see if an echocardiogram was done; if so please comment; was there a PFO? Also were the valves normal without any pathology? Echocardiogram was not performed during hospitalization; however, in a previous one the valves were normal.

This was a 64year old patient; do we know about the status of his coronaries; lipid profile? Were there any atheromatous plaques in the descending thoracic/abdominal aorta?
We do not have this information. He did not have coronary artery disease; we have also commented at our manuscript that he did not have any known medical history or comorbidities. Atheromatous plaques in aorta were not detected in computed tomography.